# *NR3C1* Glucocorticoid Receptor Gene Polymorphisms Are Associated with Membranous and IgA Nephropathies

**DOI:** 10.3390/cells10113186

**Published:** 2021-11-16

**Authors:** Michał Pac, Natalia Krata, Barbara Moszczuk, Aleksandra Wyczałkowska-Tomasik, Beata Kaleta, Bartosz Foroncewicz, Witold Rudnicki, Leszek Pączek, Krzysztof Mucha

**Affiliations:** 1Department of Immunology, Transplantology and Internal Diseases, Medical University of Warsaw, 02-006 Warsaw, Poland; michal.pac@student.wum.edu.pl (M.P.); nkrata@wum.edu.pl (N.K.); barbara.moszczuk@wum.edu.pl (B.M.); atomasik@wum.edu.pl (A.W.-T.); bartosz.foroncewicz@wum.edu.pl (B.F.); leszek.paczek@wum.edu.pl (L.P.); 2ProMix Center (ProteogenOmix in Medicine) at the Department of Immunology, Transplantology and Internal Diseases, Medical University of Warsaw, 02-006 Warsaw, Poland; 3Department of Clinical Immunology, Medical University of Warsaw, 02-006 Warsaw, Poland; beata.kaleta@wum.edu.pl; 4Computational Centre and Institute of Computer Science, University of Białystok, 15-245 Białystok, Poland; w.Rudnicki@uwb.edu.pl; 5Interdisciplinary Centre for Mathematical and Computational Modelling, University of Warsaw, 02-630 Warsaw, Poland; 6Institute of Biochemistry and Biophysics, Polish Academy of Sciences, 02-106 Warsaw, Poland

**Keywords:** genomics, glucocorticoid receptor, IgA nephropathy, membranous nephropathy, single nucleotide polymorphism

## Abstract

Glomerular diseases (GNs) are responsible for approximately 20% of chronic kidney diseases. Glucocorticoid receptor gene (*NR3C1*) single nucleotide polymorphisms (SNPs) are implicated in differences in predisposition to autoimmunity and steroid sensitivity. The aim of this study was to evaluate the frequency of the *NR3C1* SNPs—rs6198, rs41423247 and rs17209237—in 72 IgA nephropathy (IgAN) and 38 membranous nephropathy (MN) patients compared to 175 healthy controls and to correlate the effectiveness of treatment in IgAN and MN groups defined as a reduction of proteinuria <1 g/24 h after 12 months of treatment. Real-time polymerase chain reactions and SNP array-based typing were used. We found significant rs41423247 association with MN (*p* = 0.026); a significant association of rs17209237 with eGFR reduction after follow-up period in all patients with GNs (*p* = 0.021) and with the degree of proteinuria after 1 year of therapy in all patients with a glomerulopathy (*p* = 0.013) and IgAN (*p* = 0.021); and in the same groups treated with steroids (*p* = 0.021; *p* = 0.012). We also observed the association between rs41423247 and IgAN histopathologic findings (*p* = 0.012). In conclusion, our results indicate that *NR3C1* polymorphisms may influence treatment susceptibility and clinical outcome in IgAN and MN.

## 1. Introduction

Glomerular diseases (GNs) are responsible for approximately 20% of chronic kidney diseases around the world [1]. The most affected age group is young adults, among which GNs are the leading cause of end-stage renal disease (ESRD) [1]. IgA nephropathy (IgAN) and membranous nephropathy (MN) are among the dominant GNs in the adult population. Both GNs are classified as rare diseases with ORPHA numbers 761 and 97560, respectively, without targeted treatment or prognostic markers, yet. Recent years have brought new insights into a molecular background of GNs, especially in the field of -omics (genomics, transcriptomics, and proteomics) [2,3,4,5,6,7]. These discoveries revealed pathophysiologic mechanisms, which could be helpful in the development of biomarkers and targeted therapy in the future. However, none of the findings have resulted in a significant change in the treatment of GNs, and we still lack a causative therapy for any form of GN. Immunosuppressive agents, among which glucocorticoids (GCs) play an important role, are still the major therapeutic option in the GNs. Since GCs, and immunosuppressive agents in general, are associated with a wide spectrum of side effects, it becomes clear that there is an urgent need to optimize their usage for each patient individually at least until the targeted therapy is found. Although there are GN patients that benefit from steroid therapy, there is also a large unresponsive group. The reason for this is that there are interindividual differences in GC sensitivity. Several factors affecting GC sensitivity have been described [8]. However, the exact mechanism has not yet been fully explained. Among them, genetic factors with single nucleotide polymorphisms (SNP) are in the central position. According to some authors, genetic differences between patients may be responsible for up to 95% of the efficacy and side effects of drugs [9]. At present, there is no tool enabling the prediction of GC effectiveness and the occurrence of side effects, but available studies concerning steroid sensitivity may contribute to its development.

One of the described factors that could affect GC sensitivity and treatment outcomes is the *NR3C1* gene, in particular, its variants. *NR3C1* encodes intracellular steroid receptors, which mediate the molecular response to endogenous and exogenous glucocorticoids. Several SNPs of the *NR3C1* gene have been described as factors defining GC sensitivity. For example, carriers of the Asn363Ser (rs6195) and *Bcl*I (rs41423247) SNPs are known to have increased exogenous and endogenous steroid sensitivity [10,11,12,13]. Additionally, Asn363Ser carriers more often present a Cushingoid-like appearance involving increased body mass index (BMI), decreased bone mineral density (BMD) and enhanced insulin secretion after dexamethasone administration [10]. Decreased steroid sensitivity, on the other hand, was confirmed in the ER22/23EK (rs6189, rs6190) and 9ß (rs6198) SNPs carriers [13,14,15]. The ER22/23EK variant was associated with higher plasma cortisol levels and smaller reduction after dexamethasone administration, as well as low LDL cholesterol and insulin levels [14]. Major *Bcl*I polymorphism was also found to be associated with the increased risk of metabolic syndrome—a condition characterized by insulin resistance, dyslipidemia, abdominal obesity and hypertension predisposing to type 2 diabetes mellitus, cardiovascular diseases and atherosclerosis [16].

The *NR3C1* polymorphisms are implicated in differences in predisposition to certain immune-mediated diseases and in steroid treatment sensitivity. For example, ER22/23EK and 9ß SNPs carriers, in contrast to Asn363Ser and *Bcl*I carriers, have increased risk of rheumatoid arthritis (RA) [17]. Asn363Ser and *Bcl*I were also described as factors associated with lower RA activity in the postpartum period [18]. The study of pemphigus vulgaris (PV) in Chinese patients, revealed that the rs11745958 C/T and rs17209237 A/G polymorphisms may be related to decreased GC sensitivity defined as the persistence of oral or skin lesions despite therapy, and rs33388 A/T and rs7701443 A/G polymorphisms may indicate more favorable treatment outcomes and increased GC sensitivity [19]. The rs72555796 polymorphism was found to have a significant association with GC sensitivity in acute lymphoblastic leukemia [20], and the previously described rs41423247 was found to be associated with the response to prednisolone and inhaled corticosteroids in asthma [21,22].

Studies of the *NR3C1* in glomerular diseases are limited and provide contradictory results. They concern mostly the pediatric population and steroid-resistant nephrotic syndrome [23,24,25,26]. The association of the *Bcl*I, rs33389 and rs33388 *NR3C1* haplotype and increased GC sensitivity was confirmed by one study [24]. Carriers of the 9ß and TthIII-1 polymorphisms were found to have higher steroid dependence in another study [25]. However, any significant association between *NR3C1* polymorphisms was found neither in Chinese nephrotic syndrome pediatric patients [23] nor for the treatment regime in the Finnish study [26]. The only study of adults with primary nephrotic syndrome revealed the association of the *NR3C1* haplotypes containing rs10052957, rs258751 and rs6196 and steroid resistance. However, it did not include neither IgAN nor MN patients, and the study protocol did not demand kidney biopsy in any participants [27].

GC resistance is one of the important obstacles of IgAN and MN patients’ treatment. The molecular background behind the steroid ineffectiveness is poorly understood and needs to be investigated. Therefore, we aimed to evaluate the correlation between three *NR3C1* SNPs (rs6198, rs41423247 and rs17209237) and treatment effectiveness, including GC sensitivity, as well as asses the correlation between the SNPs and the long-term outcome of the diseases. To the best of our knowledge, this is the first study of this kind in IgAN and MN.

## 2. Materials and Methods

### 2.1. Ethical Approval

The study was performed following the Ethical Committee approvals of Medical University of Warsaw (no. KB/9/2010, KB/10/2010, KB/138/2010 and KB/199/2016), and all subjects provided written informed consent. The procedures followed were in accordance with the Helsinki Declaration of 1975, as revised in 2000.

### 2.2. Patients and Sample Collection

Venous blood samples were collected in tubes containing ethylene diamine tetraacetic acid (EDTA) from 38 MN and 72 IgAN kidney biopsy-proven patients of the nephrology outpatient clinic of our Department, and 175 healthy ethnically matched controls with no history of kidney disease. Furthermore, 38 MN patients were age-, gender- and ethnicity-matched in a 1:1 ratio to 38 individuals from IgAN and control groups. A summary of the characteristics of the studied groups is shown in Table 1. Among the 38 MN patients, 35 received steroids in the Ponticelli regimen, whereas out of 72 IgAN patients, 49 received steroid treatment according to KDIGO recommendations. 

The estimated glomerular filtration rate (eGFR in mL/min/1.73 m^2^) was measured using Chronic Kidney Disease Epidemiology Collaboration (CKD-EPI) equation. We checked the correlation between SNPs and outcome defined as the eGFR change divided by the follow-up time (ΔeGFR/year). The eGFR reduction of more than 1 mL/min/1.73 m^2^/year was considered as the poor outcome (fast riders, FR) and eGFR change equal or lesser than this value as considered a good outcome (slow riders, SR).

The reduction of proteinuria (g/24 h) was measured after 6 (±3) and 12 (±3) months. Patients were considered as steroid-insensitive when no reduction of < 1/24 h after 12 months of treatment was reached. We evaluated clinical outcome of IgAN and MN patients in general and in those receiving steroid treatment after the 12-month observation time. We divided patients into two groups. The first consisted of patients that reached proteinuria <1 g/day (good responders, GR) and the second of patients that did not reach that value, what was considered as a lack of effective treatment (bad responders, BR).

We obtained biopsy results assessing disease severity in light microscopy of 30/38 MN patients (Table 2) and 56/72 IgAN patients (Table 3). All 56 IgAN patients were evaluated using Hass classification and 48 individuals using Oxford classification. The full description of other patients’ biopsies was not available, as they were performed in other centers.

### 2.3. Selection of NR3C1 Polymorphisms

We used the PubMed database to find relevant literature and identify *NR3C1* and the glucocorticoid receptor pathway as key particles in the pathogenesis and treatment-modulating factors in immune-mediated diseases and possible factors determining glomerular disease and GC treatment outcomes. Three polymorphisms, rs6198, rs41423247 and rs17209237 were chosen. All of them have a minor allele frequency >5% in Caucasians, are described in the literature as possibly associated with the GC sensitivity and treatment outcomes of autoimmune diseases [8,19] and have not been studied in glomerulopathies.

### 2.4. Genotyping of the NR3C1 Gene rs6198, rs41423247 and rs17209237 Polymorphisms

Thanks to the cooperation with the Division of Nephrology in the Department of Medicine at Columbia University (New York, NY, USA), 54 IgAN and 7 MN patients and 156 controls were genome-wide genotyped using SNP microarrays (Illumina Mega Chip). Standard quality control metrics were used to process SNP data, including per-SNP and per-individual missingness rate <95%. Any SNPs with minor allele frequency <1% or Hardy Weinberg equilibrium (HWE) test *p*-value < 10^−4^ in controls were also automatically excluded.

To increase the number of studied patients, 18 IgAN and 31 MN patients and 19 controls were genotyped in our department with the use of a real-time polymerase chain reaction (RT-PCR). Genomic DNA was extracted from whole frozen blood using the TaqMan^TM^ Sample-to-SNP^TM^ Kit (Thermo Fisher Scientific, Waltham, MA, USA). The DNA concentration and purity were determined with UV spectrophotometry, measuring the 260 nm absorbance values. Genotyping was performed with the Functionally Tested Thermo Fisher Scientific assays and TaqMan^TM^ Genotyping Master Mix (Thermo Fisher Scientific, USA) by analyzing melting curves with the Applied Biosystems^TM^ 7500 Real-Time PCR system (Thermo Fisher Scientific, USA). Real-time PCR reactions were performed in 96-well PCR plates with cycling conditions optimized by Thermo Fisher Scientific.

To verify the concordance of both genotyping methods, RT-PCR genotyping was performed in a randomly selected sample of 39 participants that were previously genotyped using genome-wide SNP microarray. We detected genotype discordance for only a single SNP (rs17209237) in a single participant. RT-PCR was then repeated and provided the same result as the first RT-PCR, which was used in the downstream analysis. Thus, the genotype concordance between the genotyping was estimated at 97.4%, which we consider as acceptable for the purpose of this study.

### 2.5. Statistical Analysis

Allelic and genotypic frequencies were obtained by direct counting. The genotyping results of the controls were tested for HWE and evaluated with the χ^2^ test. The genotypic frequencies were compared among the IgAN patients, MN patients and controls using the Pearson’s chi-squared test.

Statistical analysis was performed in R. For each SNP the differences of allele composition between the following groups were examined: IgAN patients versus control group of healthy volunteers (CG), MN patient vs. CG, both diseases treated as a single group (DG) vs. control group (CG), good responders (GR) vs. bad responders (BR) and finally slow riders (SL) vs. fast riders (FR). Due to the relatively small sample size of IgAN biopsies, comparison between histopathological classes were non-significant. Therefore, we arbitrarily divided IgAN group into two categories: (1) mild phenotype (MP) comprising of I, II and III scores and severe phenotype (SP) comprising of IV and V scores in Hass scale. The analogous division was done for the Oxford classification with 0 value considered as MP, and 1 and 2 values jointly considered as SP.

For each test we built contingency table and examined independence of factors with the help of chi-squared test of independence of all factors. Low-frequency alleles were removed from the analysis.

## 3. Results

The frequency of the *NR3C1* rs6198, rs41423247 and rs17209237 polymorphism genotypes and alleles for the three groups is summarized in Table 4. The frequency of all the analyzed variants was in HWE for controls (*p* > 0.0056).

As shown in Table 4, the frequency of the heterozygous GC genotype of the *NR3C1* rs41423247 polymorphism was significantly decreased in MN patients compared both to the IgAN patients and healthy controls (*p* = 0.0261), calculated with Pearson’s chi-squared test. The analysis of the frequency of rs6198 and rs17209237 genotypes showed no significant differences among IgAN patients, MN patients and controls.

The association between rs6198, rs41423247 and rs17209237 polymorphisms and proteinuria after 12-month follow-up in the studied glomerulopathies is presented in the Table 5a–c. rs6198 and rs41423247 did not correlate with the degree of proteinuria and number of GR. Interestingly, rs17209237 correlates significantly with the therapeutical goal of proteinuria < 1 g/day in IgAN and IgAN + MN groups (*p* = 0.013, *p* = 0.021) (Table 5b,c).

It was not significant for the MN group (Table 3a). Additionally, the same SNP was significant in the same groups treated with steroids (*p* = 0.021, *p* = 0.012) (Table 6b,c). None of the differences for other SNPs reached the statistical significance.

The evaluation of ΔeGFR during follow-up, revealed that the carriers of rs17209237 AG genotype had a lesser ΔeGFR/year comparing to major AA genotype (*p* = 0.021) in a group comprised of IgAN and MN patients (Table 5c). The analysis restricted to patients on steroid treatment showed the significant association between rs17209237 AG genotype and ΔeGFR in MN as well as MN + IgAN groups (*p* = 0.026, *p* = 0.021) (Table 6a,c). The mean ΔeGFR in the studied groups for each SNP are presented in the Figure 1, Figure 2 and Figure 3 and mean changes of ΔeGFR for rs17209237 in patients treated with steroids are presented in the Figure 4. It should be noted that the follow-up of our patients was 10.19 years for all patients and 10.06 for those treated with steroids.

The analysis of correlation between *NR3C1* polymorphisms with Hass classification, showed the association between rs41423247 GC genotype and lower risk of severe phenotype in IgAN biopsies (*p* = 0.010). However, due to the number of patients, we compared GG and GC patients only. To perform test for independence for all three variants including the CC group, the validation on the larger sample size is required. The analysis of correlation with the Oxford scale, revealed the association of the GC genotype with mild phenotype in mesangial hypercellularity (M parameter) (*p* = 0.012) (Table 7). No correlation between other SNPs and histopathological changes was observed.

Furthermore, we did not find associations between SNPs and histopathological classification of MN patients.

## 4. Discussion

We decided to study NR3C1 polymorphisms in IgAN, because, although it is classified as the rare disease, it constitutes a serious problem for public health worldwide. IgAN is the most common cause of ESRD in the young adults and the most common primary glomerulonephritis in developed countries [1]. It is an immune-mediated disease with the involvement of several immunological mechanisms. Therefore, our aim was to analyze the NR3C1 SNPs of IgAN patients and to compare it with healthy controls, but also with other immune-mediated glomerulopathy, namely MN. It is the second most common glomerulonephritis found in native kidney biopsy in the adult European population [1] and the third most common in the Polish population [28]. We excluded lupus nephritis, for example, as it is a secondary glomerulonephritis, and FSGS because of its heterogeneity and plausible little involvement of immune dysregulation in the pathogenesis. Another reason for our choice was that both IgAN and MN lack targeted treatment and steroids are not effective enough.

*NR3C1* is a gene encoding glucocorticoid receptor involved in the mediation of both endogenous and exogenous glucocorticoids effects in health and disease [8]. Recent studies suggest that *NR3C1* polymorphisms could be involved in the pathogenesis of autoimmune disorders and have an impact on the GC treatment’s outcomes [17,19]. However, the role of *NR3C1* in the pathogenesis of glomerular diseases in adults have not been studied and is not fully understood. The present study was conducted to determine the frequency of rs6198, rs41423247 and rs17209237 SNPs of the *NR3C1* in the IgAN patients, MN patients compared to the healthy controls and their possible association with the disease’s prevalence, manifestation and treatment’s outcomes.

It is known that rs6198 and rs41423247 polymorphisms are associated with decreased and increased steroid sensitivity, respectively [8,10,11,14]. Currently, the loci identified by the genome-wide association study in primary MN explain 25% of the disease risk in Europeans [7]. The loci associated with the increased MN risk include genes encoding M-type phospholipase A2 receptor (*PLA2R1*) (SNP rs4664308)—the target of the autoimmune response in MN—and HLA complex class II HLA-DQ alpha-chain 1 (*HLA-DQA1*) (SNP rs2187668) [29]. A recent study of 3782 MN cases of East Asian and European ancestries identified four more risk loci [7], namely, the *PLA_2_R1* locus (SNP rs17831251); *HLA-DQA1/DRB1* locus (SNP rs9271573); and two previously undescribed loci: *NFKB1* (SNP rs230540). Although among the described loci, there are genes involved in different aspects of the immune system, including innate immunity, toll-like receptor signaling or proteasome formation. However, none of them is involved in glucocorticoid receptor signaling.

In the present study, we demonstrated that the GC genotype of the rs41423247 *NR3C1* polymorphism was less frequent in the MN patients than in healthy controls and IgAN patients, what might suggest the possible role of minor C allele in MN. The analysis of the rs6198 and rs17209237 polymorphisms did not reveal a significant difference in the frequency between the studied groups. The association between the *NR3C1* molecular network and kidney disease was previously described in a study of 16 Italian kindreds [30]. The study showed that multiple rare variants, especially in the immune-related network, including the *NR3C1* gene, may affect the susceptibility to the IgAN. Surprisingly, our results do not indicate the role of any of the studied SNPs in the predisposition to IgAN but show a significant association between the rs41423247 polymorphism and the MN occurrence. They suggest that rs41423247 might be a marker of MN. To the best of our knowledge, our study is the first to show this correlation.

The homozygous GG rs41423247 polymorphism, also known as *Bcl*I, is associated with an increased clinical glucocorticoid sensitivity demonstrated as greater total body fat content [31,32], increased cholesterol level [33] and insulin resistance [34,35]. Major *Bcl*I polymorphism was also described as the risk factor for metabolic syndrome [16]. Contradictory data can be found on the association of the rs41423247 SNP and the susceptibility to autoimmune diseases. In a study of multiple sclerosis (MS) patients, the association between the genotype containing the allele G and the higher risk of MS was described [36]. The frequency of the GG genotype was significantly higher compared to non-MS patients. On the other hand, a study of rheumatoid arthritis showed that carriers of the G allele have lower levels of baseline disease activity than non-carriers [37]. Similarly, a significantly higher frequency of G allele was described in patients with milder forms of Graves ophthalmopathy (GO) compared to patients with higher disease severity, suggesting that this SNP is associated with a lower risk of severe GO development [38]. A systematic review and meta-analysis evaluating the association between the rs41423247 SNP and autoimmune diseases, showed that the G allele might be even a protective factor for the development of systemic autoimmune disorders among Caucasian patients [39]. Therefore, the association of the G allele and a lower risk of autoimmunity or a milder disease phenotype is compliant with our results, showing that the C allele has a significantly higher frequency among MN patients than controls, which is in line with the autoimmunity in MN. These data may also suggest that polymorphisms of rs41423247 may affect immune response differently e.g., with G allele promoting low-grade inflammatory state, seen in type 2 diabetes melitus, abdominal obesity etc., and C allele promoting more specific response observed in the mentioned autoimmune diseases and including MN. Of course, this needs further exploration and larger studies on bigger and less homogenous population.

The association of the G allele of rs41423247 and steroid sensitivity in steroid-resistant nephrotic syndrome was found in one of the studies [24]. Although the association between rs6198 *NR3C1* SNP and decreased steroid sensitivity is known, its role in autoimmune diseases remains undefined. Two studies of this SNP in RA provided contradictory results. In one, the association between the 9ß variant of rs6198 SNP and RA susceptibility was described [40]. In the second, no contributing role of this polymorphism with RA was found [41]. Our results did not show any association between the rs6198 polymorphism and either IgAN or MN susceptibility. Although there is a study describing the association between the 9ß polymorphism and steroid dependence in nephrotic syndrome [25], our results did not confirm the role of this SNP in steroid effectiveness.

We found the significance of the GC genotype of rs41423247 and the mild phenotype in the Hass classification as well as M0 parameter of the Oxford classification. However, these analysis of joint Hass scores (I/II/III vs. IV/V) must be interpreted with caution, because of the sample size studied. Unfortunately, it makes impossible to unequivocally assess the risk for CC genotype that needs to be validated in another study.

The association between rs17209237 *NR3C1* and steroid sensitivity in autoimmune disorders was described in two studies. The first one found that the G allele was less frequent among GC-insensitive patients with myasthenia gravis [42], and the second one found that the A allele and AA genotype frequencies were lower in the GC-resistant group of PV patients compared to the GC-sensitive group [19]. Our results indicated the association between this SNP and therapy effectiveness in MN + IgAN and IgAN patients, defined as the proteinuria < 1 g/day after the 12 months of treatment. The same applied to the effectiveness of steroid treatment in the same groups. AA genotype was associated with lower degree of proteinuria after 1 year of treatment compared to AG genotype. The effect was stronger for patients receiving steroids what can be explained by direct action of this group of drugs on the receptor encoded by *NR3C1*. These results are congruent with the findings in PV [19].

The fact that we did not find any association between rs6198, rs41423247 or rs17209237 polymorphisms of *NR3C1* and steroid sensitivity described in other autoimmune diseases may indicate that, depending on the disorder, different molecular mechanisms determine the outcomes of GC treatment.

Our next finding was that rs17209237 AG genotype was associated with better long-term outcome of IgAN and MN defined as the ΔeGFR < 1 mL/min/1.73 m^2^/year. Such results were not observed for other studied SNPs. It can be suspected that this rs17209237 genotype may be protective against more rapid kidney function loss expressed as greater eGFR reduction. Rs17209237 is a polymorphism that was studied in two autoimmune diseases—PV and myasthenia gravis (MG) [19,42]. G allele of this polymorphism was found to be less frequent in GC-insensitive group of MG patients. The study of PV provided contradictory results with allele A and AA genotype being less frequent in GC-resistant group.

It may appear at first that the impacts of the rs17209237 on eGFR and 1-year proteinuria are contradictory. However, it is generally known that in case of glomeruli defects, the higher eGFR is, the higher proteinuria. It can be explained by greater blood volume that flows through afferent arterioles to the glomeruli what causes greater protein loss due to the defective filtration process. To evaluate the exact long-term outcome of the disease and the therapy outcomes it would be useful to assess the extension of fibrosis in glomeruli—an event caused by long-lasting proteinuria—and correlate the outcomes with eGFR loss.

We showed that AA genotype of rs17209237 correlates with decreased proteinuria, but AG genotype is associated with better long-term kidney outcome. The correlation between AA genotype and GC-sensitivity was also observed in a study of PV patients [19]. Such correlation was not observed for AG genotype in our study. This discrepancy between better treatment response—defined as proteinuria reduction for AA genotype—and better long-term kidney outcome defined as lesser eGFR loss, might be explained by unknown protective mechanism of AG genotype. Definitely, other than improvement of glomerular filtration function, which is directly associated with proteinuria rate and steroid response. This topic needs further exploration, as there are no studies on the role of *NR3C1* rs17209237 in kidney fibrosis.

## 5. Conclusions

The present study suggests an association between the NR3C1 gene rs41423247 polymorphism and MN susceptibility and severity of histopathological changes observed in IgAN kidney biopsies.

We also found the association of rs17209237 polymorphism and long-term eGFR loss and the degree of proteinuria after 1 year of steroid treatment.

The biggest limitation of our study is the sample size. On the other hand, MN and IgAN are classified as rare diseases and multiple hypotheses are the result of literature reviews, not random testing. Therefore, we are aware that the results need to be confirmed in a larger population to determine a more precise level of significance. It must be underlined that the relatively big *p*-values in our study are a result of a small sample size, not a small effect size, that is confirmed by OR values. Therefore, bearing in mind the homogenous group of patients and long-term follow-up, we believe our findings are important.

In summary, we described interesting findings about rs41423247 and rs17209237 in glomerulopathies, which should be further investigated to state final conclusions and confirm their significance.

## Figures and Tables

**Figure 1 cells-10-03186-f001:**
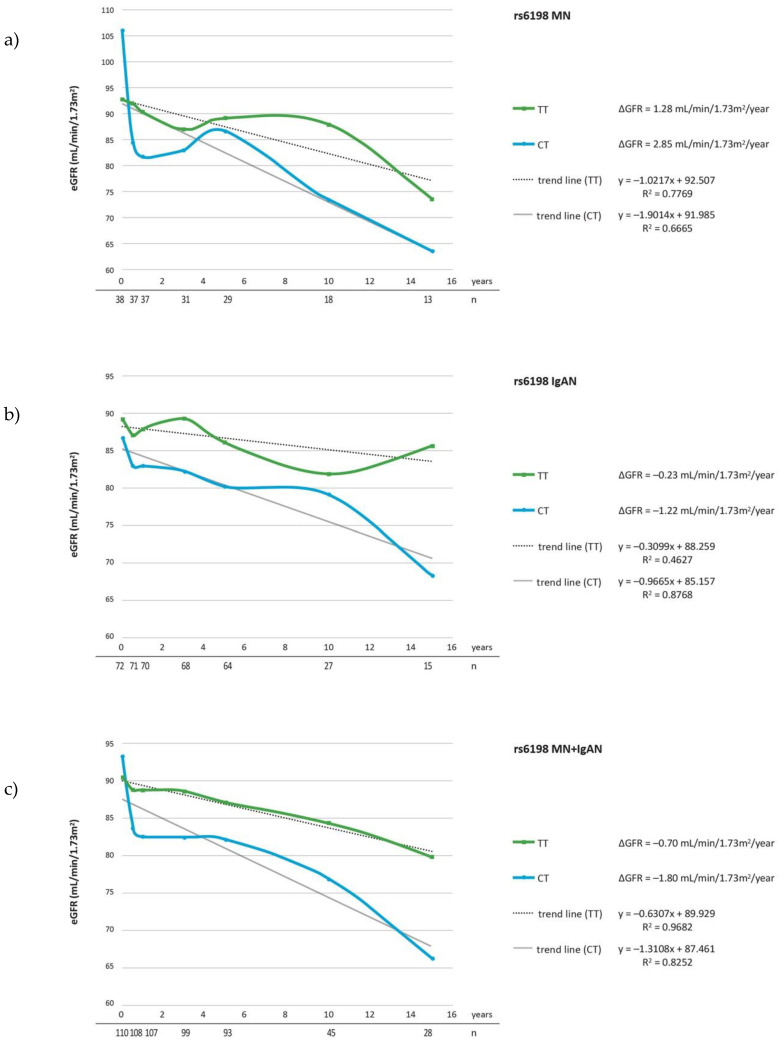
ΔeGFR for rs6198 genotypes during follow-up period in (**a**) MN patients; (**b**) IgAN patients; (**c**) MN + IgAN patients. Number of patients are expressed as (*n*).

**Figure 2 cells-10-03186-f002:**
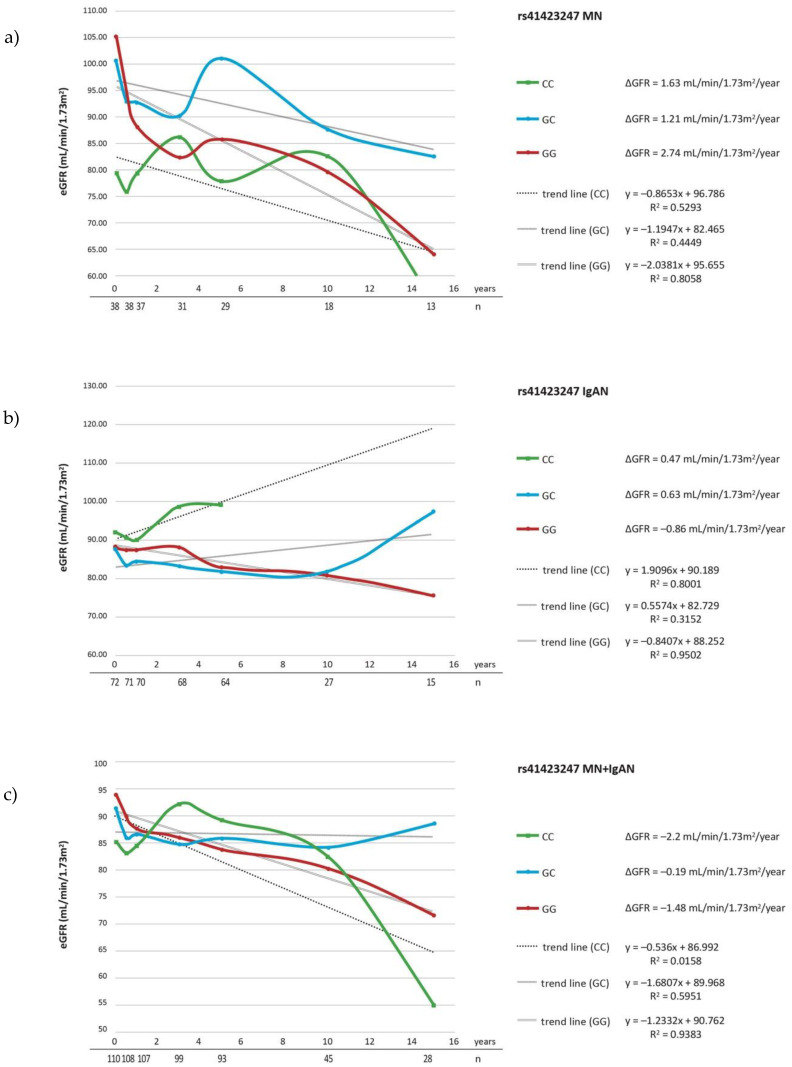
ΔeGFR for rs41423247 genotypes during follow-up period in (**a**) MN patients; (**b**) IgAN patients; (**c**) MN + IgAN patients. Number of patients are expressed as (*n*).

**Figure 3 cells-10-03186-f003:**
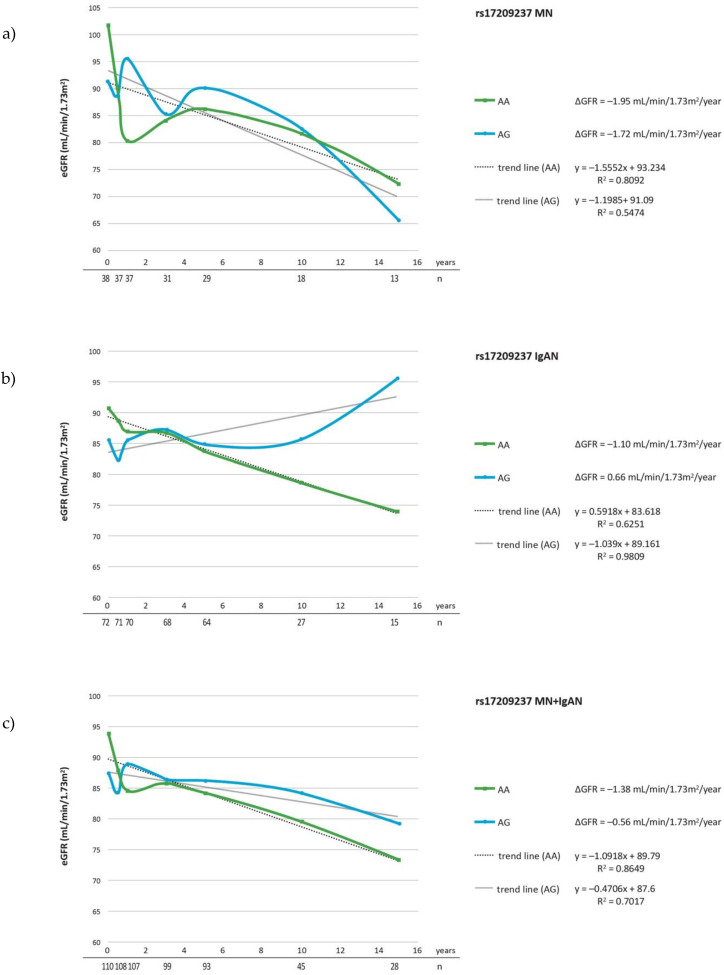
ΔeGFR for rs17209237 genotypes during follow-up period in (**a**) MN patients; (**b**) IgAN patients; (**c**) MN + IgAN patients. Number of patients are expressed as (*n*).

**Figure 4 cells-10-03186-f004:**
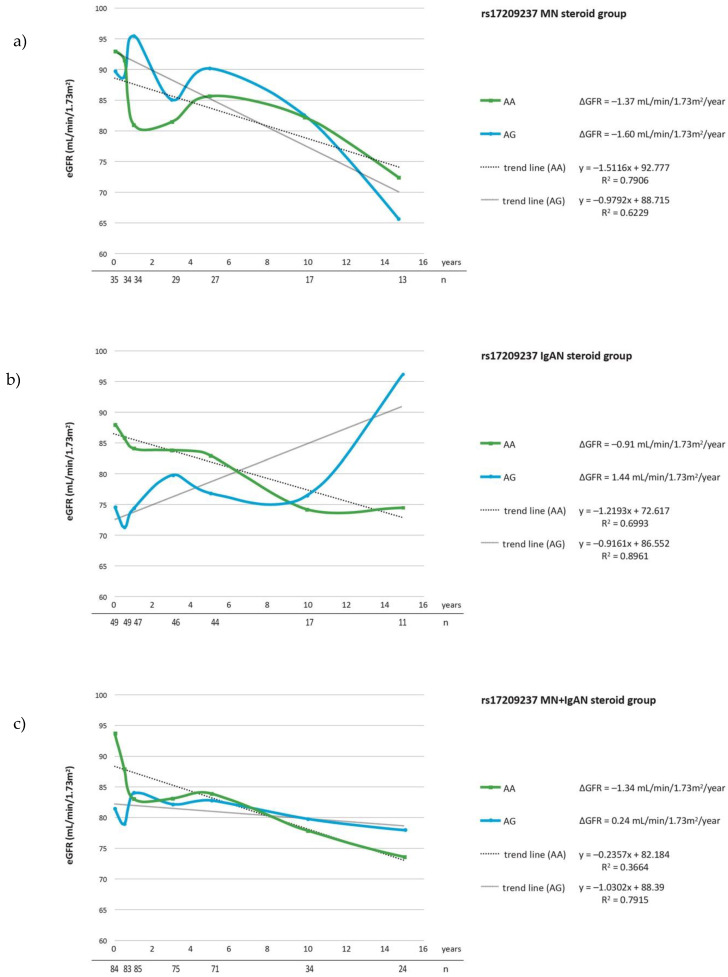
ΔeGFR for rs17209237 genotypes during follow-up period in (**a**) MN patients treated with steroids; (**b**) IgAN patients treated with steorids; (**c**) MN + IgAN patients treated with steroids. Number of patients are expressed as (*n*).

**Table 1 cells-10-03186-t001:** Characteristics of IgAN and MN patients, and healthy controls. Blood samples were collected for NR3C1 polymorphism genotyping. Number of patients (*n*). ACE-I—angiotensin-converting-enzyme inhibitors; ARB—angiotensin II receptor blockers; AZA—azathioprine; CsA—cykclosporine; CYF—cyclophosphamide; MMF—mycophenolate mofetil; TAC—tacrolimus.

Parameter	IgAN *n* = 72	MN *n* = 38	Controls *n* = 175
Age, year	33.96 ± 12.03	42.89 ± 14.37	48.7 ± 17.9
Gender (male/female, *n*)	38/34	24/14	86/89
Creatinine (mg/dL)	1.09 ± 0.48	0.92 ± 0.45	0.93 ± 0.20
GFR (mL/min/1.73 m^2^)	88.38 ± 30.98	96.89 ± 29.31	85.84 ± 19.30
24-h urine protein excretion, mg/d	1.73 ± 2.07	4.44 ± 3.37	-
ACE-I (*n*)	44	26	-
ARB (*n*)	19	8	-
CsA (*n*)	2	7	-
TAC (*n*)	0	1	-
CYF (*n*)	0	4	-
AZA (*n*)	4	1	-
MMF (*n*)	1	0	-

**Table 2 cells-10-03186-t002:** MN stages (light microscopy). Number of patients (*n*).

Stages *n* = 30	
I	8
I/II	4
II	13
II/III	2
III	3
IV	0

**Table 3 cells-10-03186-t003:** Combined Hass and Oxford classifications of IgAN. Number of patients (*n*).

Oxford Classification *n* = 48	Hass Classification *n* = 56				
	I	II	III	IV	V
	*n* = 1	*n* = 5	*n* = 25	*n* = 18	*n* = 7
M					
0	1	5	20	1	3
1	0	0	0	14	4
E					
0	1	5	11	12	4
1	0	0	9	3	3
S					
0	1	1	8	1	0
1	0	4	12	14	7
T					
0	1	4	18	11	1
1	0	1	2	4	3
2	0	0	0	0	3
C					
0	1	5	13	9	5
1	0	0	7	6	2
2	0	0	0	0	0

**Table 4 cells-10-03186-t004:** Frequencies of rs6198, rs41423247 and rs17209237 polymorphisms in IgAN, MN and control groups.

Genotypes						
**rs6198**						
	CC		CT		TT	
**IgAN + MN**	0	0%	36	33%	74	67%
**MN**	0	0%	12	32%	26	68%
**IgAN**	0	0%	24	33%	48	67%
**Control**	2	1%	48	27%	125	71%
**rs41423247**						
	CC		GC		GG	
**IgAN + MN**	19	17%	44	40%	47	43%
**MN**	10	26%	12	32%	16	42%
**IgAN**	9	13%	32	44%	31	43%
**Control**	23	13%	78	45%	74	42%
	OR = 0.349, *p* = 0.026					
**rs17209237**						
	GG		AG		AA	
**IgAN + MN**	1	1%	42	42%	56	57%
**MN**	2	5%	16	42%	20	38%
**IgAN**	0	0%	30	42%	42	58%
**Control**	2	1%	58	33%	115	66%

**Table 5 cells-10-03186-t005:** The correlation between rs6198, rs41423247 or rs17209237 NR3C1 polymorphisms with ΔeGFR/year or proteinuria (g/day) after 1-year follow-up time for (**a**) MN; (**b**) IgAN; (**c**) all patients. Number of patients (*n*).

**(a)**	**MN**												
**SNP**	**Genotype**	**ΔeGFR** **≥ −1 per Year (*n*)**	**ΔeGFR** **< −1 per Year (*n*)**	**RR**	**OR**	**Chi^2^**	** *p* **	**Proteinuria** **< 1 g/day (*n*)**	**Proteinuria** **≥ 1 g/day (*n*)**	**RR**	**OR**	**Chi^2^**	** *p* **
**rs6198**	CT	5	7	n.s.	n.s.	0.433	0.510	7	5	n.s.	n.s.	0.175	0.675
	TT	8	18	n.s.	n.s.			17	9	n.s.	n.s.		
**rs41423247**	CC	4	6	n.s.	n.s.	1.9	0.388	7	3	n.s.	n.s.	0.324	0.850
	GC	6	6	n.s.	n.s.			7	5	n.s.	n.s.		
	GG	4	12	n.s.	n.s.			10	6	n.s.	n.s.		
**rs17209237**	AA	4	16	1.6	4	3.6	0.058	13	7	0.8	0.69	0.286	0.593
	AG	8	8					9	7				
**(b)**	**IgAN**												
**SNP**	**Genotype**	**ΔeGFR** **≥ −1 per Year (*n*)**	**ΔeGFR** **< −1 per Year (*n*)**	**RR**	**OR**	**Chi^2^**	** *p* **	**Proteinuria** **< 1 g/day (*n*)**	**Proteinuria** **≥ 1 g/day (*n*)**	**RR**	**OR**	**Chi^2^**	** *p* **
**rs6198**	CT	8	16	n.s.	n.s.	2.266	0.132	18	5	n.s.	n.s.	0.156	0.693
	TT	25	23	n.s.	n.s.			34	12	n.s.	n.s.		
**rs41423247**	CC	5	4	n.s.	n.s.	0.546	0.761	8	1	n.s.	n.s.	4.251	0.119
	GC	15	17	n.s.	n.s.			19	11	n.s.	n.s.		
	GG	13	18	n.s.	n.s.			25	5	n.s.	n.s.		
**rs17209237**	AA	16	26	1.429	2.125	2.431	0.119	36	6	0.351	0.242	6.195	0.013
	AG	17	13					16	11				
**(c)**	**All Patients**												
**SNP**	Genotype	**ΔeGFR** **≥ −1 per Year (*n*)**	**ΔeGFR** **< −1 per Year (*n*)**	**RR**	**OR**	**Chi^2^**	** *p* **	**Proteinuria** **< 1 g/day (*n*)**	**Proteinuria** **≥ 1 g/day (*n*)**	**RR**	**OR**	**Chi^2^**	** *p* **
**rs6198**	CT	13	23	n.s.	n.s.	0.716	0.397	25	10	n.s.	n.s.	0.004	0.095
	TT	33	41	n.s.	n.s.			51	21	n.s.	n.s.		
**rs41423247**	CC	8	11	n.s.	n.s.	1.248	0.536	15	4	n.s.	n.s.	2.85	0.241
	GC	21	33	n.s.	n.s.			26	16	n.s.	n.s.		
	GG	17	30	n.s.	n.s.			35	11	n.s.	n.s.		
**rs17209237**	AA	20	42	1.484	2.5	5.302	0.021	49	13	0.501	0.369	5.33	0.021
	AG	25	21					25	18				
	GG	1	1					2	0				

**Table 6 cells-10-03186-t006:** The correlation between rs17209237 NR3C1 polymorphisms with ΔeGFR (mL/min/1.73 m^2^/year) or proteinuria (g/day) after 1-year follow-up time in patients receiving steroid treatment for (**a**) MN; (**b**) IgAN; (**c**) all. Number of patients (*n*).

(**a**)	**MN**												
**SNP**	**Genotype**	**ΔeGFR** **≥ −1 per Year (*n*)**	**ΔeGFR** **< −1 per Year (*n*)**	**RR**	**OR**	**Chi^2^**	** *p* **	**Proteinuria** **< 1 g/day (*n*)**	**Proteinuria** **≥ 1 g/day (*n*)**	**RR**	**OR**	**Chi^2^**	** *p* **
**rs17209237**	AA	3	15	1.79	5.714	4.95	0.026	12	6	0.714	0.571	0.609	0.435
	AG	8	7					8	7				
	GG	1	1					2	0				
(**b**)	**IgAN**												
**SNP**	**Genotype**	**ΔeGFR** **≥ −1 per Year (*n*)**	**ΔeGFR** **< −1 per Year (*n*)**	**RR**	**OR**	**Chi^2^**	** *p* **	**Proteinuria** **< 1 g/day (*n*)**	**Proteinuria** **≥ 1 g/day (*n*)**	**RR**	**OR**	**Chi^2^**	** *p* **
**rs17209237**	AA	10	21	1.4	2.1	1.510	0.219	21	6	0.404	0.23	5.347	0.021
	AG	9	9					9	11				
	GG	0	0					0	0				
(**c**)	**All patients**												
**SNP**	**Genotype**	**ΔeGFR** **≥ −1 per Year (*n*)**	**ΔeGFR** **< −1 per Year (*n*)**	**RR**	**OR**	**Chi^2^**	** *p* **	**Proteinuria** **< 1 g/day (*n*)**	**Proteinuria** **≥ 1 g/day (*n*)**	**RR**	**OR**	**Chi^2^**	** *p* **
**rs17209237**	AA	13	36	1.515	2.94	5.306	0.021	39	10	0.435	0.291	6.355	0.012
	AG	17	16					17	15				
	GG	1	1					2	0				

**Table 7 cells-10-03186-t007:** Association between rs41423247 and Oxford classification of M parameter in IgAN kidney biopsies. Number of patients (*n*). M—mesangial hypercellularity in Oxford classification.

Oxford Classification *n* = 44						
SNP	Genotype	M0	M1	OR	Chi^2^	*p*
**rs41423247**	GG	10	12	**5.40**	**6.29**	**0.012**
	GC	18	4			

## Data Availability

The data presented in this study are available on request from the corresponding author. The data are not publicly available due to confidential reasons.

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
