# Peer review of "NR3C1 Glucocorticoid Receptor Gene Polymorphisms Are Associated with Membranous and IgA Nephropathies"

_cells, 2021, doi:10.3390/cells10113186_

Round 1

Reviewer 1 Report

Explore the steroid sensitivity is an important factor for treatment outcome and its side effect in longstanding therapy in glomerulonephritis. It is novel findings for treatment outcome based on glucocorticoid receptor gene polymorphism.

Major points

MN treated with Ponticelli regimen, and IgAN with KDIGO recommendation showed as patient’s sample collection. Results that author depicted were focusing on glucocorticoid receptor gene polymorphism and eGFR mainly.

Presumably many readers would like to know about prognosis of MN and IgAN based on their histological evaluation on biopsy. Disease intensity should be considered by Oxford classification in IgAN, stage criteria by electron microscopy in MN.

  • Author should describe patient’s characteristics in their histological grade added in table 1.
  • Author should also describe the reason for the choice of disease (MN and IgAN) in discussion.
  • Concomitant use of immunosuppression, ARB/ACEI is also needed to describe.

Reviewer 2 Report

The aim of this article is to evaluate the correlation between NR3C1 SNPs and treatment effectiveness and long term outcome of IgAN and MN. This is not an original paper since polymorphisms of NRC3C1 gene has been previously studied in children with nephrotic syndrome.  However, NR3C1 SNPs in adults has not been determined.  Although the experimental design is adequate, there are several concerns in the presentation and the interpretation of the results.    Following are some specific concerns about the manuscript:

Major concerns:

  1. The major concern is the use of two different methods to determine NR3C1 SNPs. How accurate is each method?  The polymorphisms detected by both methods were verified by counting target and control reads containing the polymorphisms?
  2. Usually patients with steroids demonstrate kidney benefit as evidenced by slow decline in eGFR and lower urinary protein excretion. The authors showed decreased proteinuria in rs7209237 NR3C1 AA genotype but better kidney long-term outcome in the AG genotype.  Additional comments about this discrepancy should be provided.

Minor concerns:

  1. There are some spelling mistakes: Lines 193 Rs6198 instead of rs6198

Round 2

Reviewer 1 Report

This manuscript comes along, precisely corrected by reviewer's comment.